# Learn From One Specialized Sub-Teacher: One-to-One Mapping for Feature-Based Knowledge Distillation

**Khouloud Saadi, Jelena Mitrović, Michael Granitzer**
University of Passau, Germany
{Khouloud.Saadi, Jelena.Mitrovic, Michael.Granitzer}@uni-passau.de

## Abstract

Knowledge distillation is known as an effective technique for compressing over-parameterized language models. In this work, we propose to break down the global feature distillation task into $N$ local sub-tasks. In this new framework, we consider each neuron in the last hidden layer of the teacher network as a specialized sub-teacher. We also consider each neuron in the last hidden layer of the student network as a focused sub-student. We make each focused sub-student learn from one corresponding specialized sub-teacher and ignore the others. This will facilitate the task for the sub-student and keep it focused. Our proposed method is novel and can be combined with other distillation techniques. Empirical results show that our proposed approach outperforms the state-of-the-art methods by maintaining higher performance on most benchmark datasets. Furthermore, we propose a randomized variant of our approach, called Masked One-to-One Mapping. Rather than learning all the $N$ sub-tasks simultaneously, we focus on learning a subset of these sub-tasks at each optimization step. This variant enables the student to digest the received flow of knowledge more effectively and yields superior results.

## 1 Introduction

Large language models, also known as general purpose language models, have revolutionized the NLP domain (Devlin et al., 2019; Brown et al., 2020; Radford et al., 2019; Clark et al., 2020). They are large architectures composed of several transformer blocks (Vaswani et al., 2017), typically trained on large unlabeled corpora in a self-supervised way (Devlin et al., 2019; Brown et al., 2020). They achieved state-of-the-art (SOTA) performance on downstream tasks through fine-tuning when the data is scarce (Devlin et al., 2019; Brown et al., 2020). However, since these models are typically large and computationally expensive, such as BERT with millions of parameters (Devlin et al., 2019) and GPT-3 with billions of parameters (Brown et al., 2020), they are not highly adapted to real-world applications like mobile computing. Model compression is an active field of research that focuses on effectively reducing the model size without significant performance degradation (Xu and McAuley, 2022; Frankle and Carbin, 2018; Jiao et al., 2020).

Knowledge Distillation (KD) by (Hinton et al., 2015) is one of the effective compression techniques in NLP where the knowledge of a highly capable large model, i.e., teacher, is transferred to a smaller efficient model, i.e., student. KD essentially requires designing a loss function to minimize the distance of the output or the intermediate representations between the student and the teacher (Sanh et al., 2019; Jiao et al., 2020; Hinton et al., 2015). To distill the intermediate representations, previous research relied on the mean square error (MSE) as an objective function between the student and the teacher global representations (Sun et al., 2019; Jiao et al., 2020). However, this metric is sensitive to scale (Saadi and Taimoor Khan, 2022) and it is not accurate in high dimensional space (Aggarwal et al., 2001; Houle et al., 2010). Other works used the cosine distance as an alternative, but, it also has several limitations (Zhou et al., 2022; Schütze et al., 2008) such as not performing well with sparse data.

In this work, we focus on distilling the teacher's last hidden layer representation into the student's last hidden layer representation. To achieve that, we propose a novel KD approach where we reformulate this feature distillation task from a global problem to $N$ local sub-problems. Here, the term "local" refers to the consideration of each dimension or neuron individually. In this work, our main assumption is that the student last hidden layer has the same dimension as the teacher last hidden layer. We note that this is a common assumption in feature distillation (Sanh et al., 2019).

In our proposed framework, we consider each neuron in the teacher's last hidden layer as a specialized sub-teacher and each neuron in the student's last hidden layer as a focused sub-student. Each specialized sub-teacher is in charge of distilling its knowledge to the corresponding focused sub-student. We call this a one-to-one matching between the teacher and the student last hidden layers units. To accomplish this one-to-one distillation, we propose a novel objective function where we maximize the per-batch correlation between the outputs of each specialized sub-teacher and its corresponding focused sub-student. Empirical results show that studying the global feature distillation task from a local viewpoint helped the student to meet the global teacher's features representation. In addition, we propose an augmented variant of our approach where the model learns a subset of sub-tasks at a time. To sum up, our contributions are the following:

- We reformulate the global feature distillation task into $N$ local sub-tasks where we do a one-to-one mapping between the last hidden layers units of the teacher and the student models.

- We propose a local correlation-based objective function to achieve the One-to-One Mapping distillation.

- We conduct extensive experiments: 1. Standalone experiments on the SQUAD V1 and the IMDB dataset. 2. Comparison with the competing methods on 8 GLUE datasets (Wang et al., 2018). Our approach performs the best in most cases.

- We present an augmented variant of our proposed approach called: Masked One-to-One Mapping. Empirical results show the effectiveness of this new variant.

## 2 Related Work

Model compression techniques are employed to effectively reduce the size of a neural network while maintaining good performance. Various approaches have been proposed such as pruning (Frankle and Carbin, 2018). It aims to remove unimportant structures, e.g., weights, neurons, and even entire layers from the model (Lagunas et al., 2021; Prasanna et al., 2020). Another technique is weight sharing (Long et al., 2017), where different parts of the model use the same set of weights

to perform computations (Lan et al., 2019; Reid et al., 2021). Quantization (Zhou et al., 2017) is yet another strategy used in model compression, where the weights and activations in the network are represented using lower bit integers instead of higher precision floating-point numbers (Kim et al., 2021; Prato et al., 2020).

Knowledge distillation (Hinton et al., 2015; Zhang et al., 2020; Sanh et al., 2019), which is the focus of this paper, is also a successful compression technique. It involves training a smaller, more efficient student model to mimic the behavior of a larger, knowledgeable teacher model. By leveraging the knowledge of the teacher model to the student model, the student can achieve comparable performance to the teacher while maintaining a lower size. Knowledge distillation has been proven to be an effective technique for large language model compression (Sun et al., 2019; Sanh et al., 2019). It can be applied during the pre-training stage to generate general-purpose distilled models (Jiao et al., 2020; Sanh et al., 2019) and during the fine-tuning stage to generate task-specific distilled models (Zhou et al., 2021; Liang et al., 2020; Sun et al., 2019).

In (Kovaleva et al., 2019), the authors show that large language models, e.g., BERT, suffer from over-parametrization in domain-specific tasks. Thus, previous work has been improving the task-specific distillation. Several methods focused on enhancing the objectives of the distillation process. These improvements mainly focused on which part of the teacher architecture can be distilled into the student architecture such as the attention matrices (Jiao et al., 2020), the different hidden states (Sun et al., 2019), and the prediction layer (Hinton et al., 2015).

Coming up with effective objective functions to distill the knowledge from any part of the teacher into the student is critical. For the logit-based KD, the KL divergence or the MSE are used as objective functions to minimize the distance between the logits of the student and the logits of the teacher (Hinton et al., 2015; Zhou et al., 2021). Following that, in (Zhao et al., 2022), the authors provided a novel viewpoint to study the logit distillation by reformulating the classical KL divergence loss into two parts, which showed a good improvement. In the feature-based KD, the MSE and the cosine distance are mainly used as objective functions to align the intermediate layers representations of the

teacher and the student (Sun et al., 2019; Jiao et al., 2020; Sanh et al., 2019).

In this work, we argue that relying on the MSE and the cosine distance as loss functions to align the intermediate representations between the student and the teacher is not an optimal choice. MSE is sensitive to scale and does not perform well in high-dimensional space. In a high dimensional space, which is the case in a neural network, the data tend to be sparse and all the data points become uniformly distant from each other (Aggarwal et al., 2001; Houle et al., 2010). In fact, the embeddings (features representation) provided by BERT-like models are high-dimensional and sparse tensors in nature (Li et al., 2022). Thus, using MSE or cosine distance to measure the distance between these sparse representations is not optimal as explained and advocated in (Aggarwal et al., 2001). So, splitting the task into sub-tasks and using one-to-one metric alleviates this problem and breaks the curse of dimensionality. In (Zhou et al., 2022), it is also shown that the cosine distance is not an accurate measurement of similarity between BERT embeddings. Moreover, the cosine distance measure can be also affected by sparsity. In fact, in high-dimensional space, it can output large angles between two sparse vectors although they are similar in the non-zero components (Schütze et al., 2008). Another important limitation of the cosine distance is that it is a global metric and sensitive to unit permutation. For example, let $W_1$ be the output tensor of a given layer in the teacher and $W_2$ be the output of the corresponding layer in the student. In the following example, we compute the cosine distance (CosD) from $W_1$ and $W_2$:

$$W_1 = [1, 5]$$

$$W_2 = [5, 1]$$

$$CosD(W_1, W_2) = 1 - \frac{W_1 \cdot W_2}{\|W_1\|_2 \|W_2\|_2} = 0.6154$$

It is a high value but the layers actually learned the same representation.

In our work, instead of treating feature distillation as a global task (Sanh et al., 2019; Sun et al., 2019) between teacher-student layers, we reformulate it as multiple local sub-tasks. Furthermore, for each per-dimension KD sub-task, instead of using MSE or cosine distance (Sanh et al., 2019; Jiao et al., 2020; Sun et al., 2019), we propose a noval loss function that utilizes a per-batch correlation function. This improves the student's ability

to meet the global representation provided by the teacher. We note that the initial results of our work on the one-on-one mapping approach are presented in (Saadi et al., 2023).

## 3 Our Approach: One-to-One Mapping

In this work, we provide a new viewpoint on how to study the feature distillation task. We break down the global KD task into multiple local KD sub-tasks. Specifically, in the last hidden layer of the teacher, we consider each unit as a specialized sub-teacher. In the last hidden layer of the student, we also treat each unit as a focused sub-student. Each sub-student should concentrate and learn only from the corresponding specialized sub-teacher. We refer to this as One-to-One Mapping, which is reformulated as the cross-correlation between the outputs of each specialized sub-teacher and its corresponding focused sub-student. The mapping between each sub-student and each sub-teacher is fixed, i.e. based on an index, and thus remains consistent over batches and epochs. Although our approach can be applied to different student-teacher hidden layers, as in (Sun et al., 2019), this work specifically focuses on the last hidden layer.

Typically, in a KD framework, as illustrated in Figure 1, we have the teacher network modeled by $f_\theta$, which is an over-parameterized knowledgeable model. The student network is modeled by $f'_{\theta'}$ which is an efficient model that has a lower number of parameters compared to the teacher. The input batch $X$ is fed to $f_\theta$ and $f'_{\theta'}$ simultaneously to produce the batches of features representation $\mathbf{Y}_t$ and $\mathbf{Y}_s$, respectively.

We assume that $\mathbf{Y}_t$ and $\mathbf{Y}_s$ are the features representation of the last hidden layer of the teacher $\mathbf{h}_t^{Last}$ and the last hidden layer of the student $\mathbf{h}_s^{Last}$, respectively. $\mathbf{Y}_s$ and $\mathbf{Y}_t$ are assumed to be mean-centered over the batch dimension. We assume that $\mathbf{h}_t^{Last}$ and $\mathbf{h}_s^{Last}$ have $N$ hidden units. The $\text{unit}_i$ in $\mathbf{h}_t^{Last}$ represents the specialized sub-teacher$_i$ and the $\text{unit}_i$ in $\mathbf{h}_s^{Last}$ represents the focused sub-student$_i$. The sub-task$_i$ is distilling the Knowledge from the specialized sub-teacher$_i$ to the focused sub-student$_i$ by reducing the distance between the features learned by each of them.

To simplify the task for the sub-student$_i$ and to keep it focused, we force it to learn only from the sub-teacher$_i$ and ignore the other sub-teachers. We reformulate the objective function in this one-to-one mapping as maximizing the cross-correlation

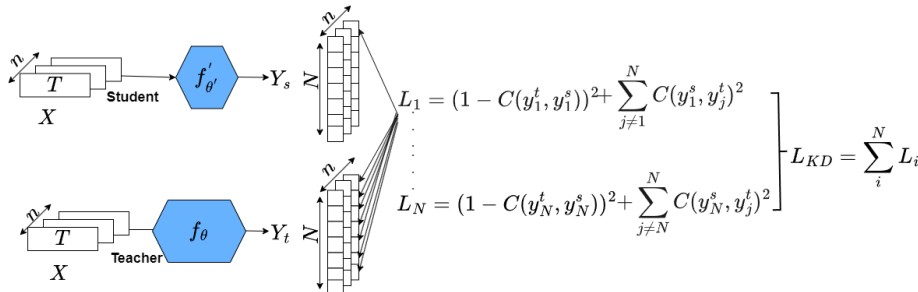

Figure 1: One-to-One Mapping: $X$ is the input batch. $n$ is the batch size. T is a given sample. C is the cross correlation function. The **Student** and the **Teacher** are modeled by $f'_{\theta'}$ and $f_\theta$, respectively. $Y_s$ and $Y_t$ are the features representation of the last hidden layer of the student and the teacher, respectively.

between the two variables $y_i^t$ and $y_i^s$. $y_i^t$ and $y_i^s$ are the output values of the sub-teacher$_i$ and the sub-student$_i$, respectively. The variables $y_i^t$ and $y_i^s$ have $n$ samples coming from the different examples in the input batch. Maximizing the correlation across batches between the two aforementioned variables, i.e., minimizing the following loss function:

$$l_i = (1 - C_{ii})^2 \qquad (1)$$

where $C_{ii}$ is the cross-correlation value between the variables $y_i^t$ and $y_i^s$:

$$C_{ii} = \frac{\sum_{b=1}^n y_{b,i}^t y_{b,i}^s}{\sqrt{\sum_{b=1}^n (y_{b,i}^t)^2}\sqrt{\sum_{b=1}^n (y_{b,i}^s)^2}} \qquad (2)$$

$b$ is the index of a given sample in the input batch $X$. $n$ is the number of examples in the input batch $X$. $i, j$ index the output dimension of the last hidden layer in both the teacher and the student. In fact, $i$ is the index of the $i_{th}$ element in the output and it is also the index of the $i_{th}$ neuron in the last hidden layer, i.e., sub-teacher$_i$ or sub-student$_i$.

As we want to make the task easier for the sub-student$_i$, so it can effectively digest the received information, we force it to mimic only the teacher$_i$ and repel all knowledge coming from the other teachers. This results in minimizing the following term:

$$R_i = \sum_{j \neq i}^N C_{ij}^2 \qquad (3)$$

Thus, our final per-dimension KD loss function, as shown in Figure 1, is:

$$L_i = l_i + R_i = (1 - C_{ii})^2 + \sum_{j \neq i}^N C_{ij}^2 \qquad (4)$$

In $L_i$, the first term is for maximizing the cross-correlation over batches between the output of the

sub-student$_i$ and the output of the sub-teacher$_i$. The second term is for minimizing the cross-correlation between the sub-student$_i$ and each sub-teacher$_j$ given $j \neq i$ and $j \in \{1, 2, ..., N\}$. Our end distillation loss is the sum of the $N$ local KD losses.

$$L_{KD} = \sum_i^N L_i = \sum_i^N (1 - C_{ii})^2 + \sum_i^N \sum_{j \neq i}^N C_{ij}^2 \qquad (5)$$

Additionally, we introduce $\lambda_1$ and $\lambda_2$ as the weights to control the contribution of the first term and the second term of the loss function, respectively:

$$L_{KD} = \lambda_1 \sum_i^N (1 - C_{ii})^2 + \lambda_2 \sum_i^N \sum_{j \neq i}^N C_{ij}^2 \qquad (6)$$

The final training loss of the original student is:

$$L = \alpha L_{CE} + \beta L_{KD} \qquad (7)$$

Where $L_{CE}$ is the classical cross entropy loss between the student predictions and the ground truth labels:

$$L_{CE} = -\frac{1}{n} \sum_i^n y_i \times \log f'_{\theta'}(x_i) \qquad (8)$$

Where $x_i$ is an input sample and $y_i$ is its ground truth label. $n$ is the number of samples per batch.

Empirical results will show that our One-to-One Mapping approach effectively facilitates the alignment of the student features representation with the global teacher feature representation. Our approach can be implemented at a low computational cost and can be combined with other knowledge distillation methods.

## 4 Experimental Results

In our experiments, the teacher is the BERT-base model, with 110 million parameters, after being fine-tuned on each of the datasets for 3 epochs. The student is the DistilBERT-base with 66 million parameters. $N$, which is the number of neurons, in the last hidden layer of the teacher and the student, is equal to 768. All experiments are repeated for 5 random seeds, the learning-rate is set to 5e-5, the maximum sequence length is set to 128, and the batch size is set to 16.

### 4.1 Stand-Alone Experiments

#### 4.1.1 Stand-Alone Results

In this stand-alone performance evaluation, we experiment with the Stanford Question Answering Dataset (SQuAD-V1) (Rajpurkar et al., 2016) and the Internet Movie Database dataset (IMDB) (Maas et al., 2011). The SQUAD-V1 is a reading comprehension dataset, consisting of questions posed by crowd workers on a set of Wikipedia articles, where the answer to every question is a segment of text, or span, from the corresponding reading passage. The reported metrics for SQUAD are the Exact Match (EM) (Rajpurkar et al., 2016), which measures the percentage of predictions that match any one of the ground truth answers exactly, and the F1-score (Rajpurkar et al., 2016), which measures the average overlap between the prediction and ground truth answer. The IMDB dataset is a sentiment analysis dataset consisting of 50,000 movie reviews labeled as positive or negative.

We distill the last hidden layer representation of the teacher into the last hidden layer of the student. We add our designed KD loss, the MSE as in (Sun et al., 2019), and the cosine distance as in (Sanh et al., 2019), as stand-alone regularizers to the hard loss between the student predictions and the ground truth labels. This will show the effectiveness of our proposed objective function for the feature distillation task. We experiment with 3 and 10 epochs. The weight of each KD stand-alone loss and the weight of the hard loss are fixed to 0.5 (Sanh et al., 2019; Jiao et al., 2020).

In Table 1, $BERT_{12}$ and $Distilbert_6$ refer to the BERT-base pre-trained language model with 12 transformer blocks and to the DistilBERT-base model with 6 transformer blocks, respectively. $Distilbert_6$-FT stands for fine-tuning the student without any distillation. $Distilbert_6$-cosD stands for feature distillation with cosine distance objec-tive function. $Distilbert_6$-MSE stands for feature distillation with MSE loss.

As shown in Table 1, our proposed method achieves the best results on the feature distillation task. It could outperform MSE and cosine distance on the squad and the imdb datasets. The results show that our approach performs the best when the distillation task is run for 3 epochs. It achieves an Exact Match (EM) score of 78.79% and an F1 score of 86.95% on the squad dataset. It also gives an accuracy of 93.87% on the imdb dataset. In Table 2, For 10 epochs, while the performance of the other approaches decreased, ours effectively increased. It achieves 79.46% and 87.48% as EM and F1, respectively, on the squad dataset and 93.96% as accuracy on the imdb dataset. This proves that our approach yields better representation of the student features. Another noteworthy aspect is that the standard deviation of the results of our approach is low compared to the others. This indicates the stability and consistency of our proposed method.

#### 4.1.2 Sensitivity Analysis

In this subsection, we investigate the impact of the weight value assigned to each stand-alone loss on the performance of the student model. As a reminder, in this stand-alone evaluation, the overall loss function for the student model is computed as the weighted sum of the hard loss and each stand-alone KD loss. The KD stand-alone loss is computed between the last hidden layer of the student and the last hidden layer of the teacher. As KD stand-alone regularizers, we will compare the MSE, the cosine distance, and our newly formulated loss function. We maintain a fixed weight for the hard loss, i.e., 0.5, while varying the weight assigned to each stand-alone loss term. Here, the number of epochs is set to 3.

As illustrated in Figure 2, on the IMDB dataset, the accuracy of the student model utilizing our stand-alone loss consistently surpasses those employing MSE and cosine distance for all weight values. Additionally, as shown in Figure 3, on the squad dataset, utilizing our novel loss consistently yields the highest performance in terms of both the Exact Match (EM) and F1 metrics, across all hyper-parameter values, when compared to MSE and cosine distance. This highlights the robustness and effectiveness of our novel approach.

| Approach | SQUAD-V1 (%) | IMDB (%) |
|---|---|---|
| BERT$_{12}$ (teacher) | 80.36/88.13 | 94.06 |
| Distilbert$_6$-FT | 77.43±0.22/85.67±0.10 | 93.19±0.09 |
| Distilbert$_6$-cosD | 77.82±0.29/85.92±0.23 | 93.66±0.08 |
| Distilbert$_6$-MSE | 77.72±0.21/85.86±0.12 | 93.49±0.08 |
| **Distilbert$_6$-ours** | **78.79±0.12/86.95±0.06** | **93.87±0.01** |

Table 1: Stand-Alone regularizers. SQUAD-V1: The evaluation is reported as Exact Match (EM) and F1 on the dev set. IMDB: The evaluation is reported as accuracy on the test set. Results are the average and the standard deviation of 5 random seeds. Training for 3 epochs.

| Approach | SQUAD-V1 (%) | IMDB (%) |
|---|---|---|
| BERT$_{12}$ (teacher) | 80.36/88.13 | 94.06 |
| Distilbert$_6$-FT | 74.32±0.37/83.69±0.31 | 92.71±0.08 |
| Distilbert$_6$-cosD | 75.78±0.27/84.57±0.11 | 93.90±0.08 |
| Distilbert$_6$-MSE | 75.17±0.18/84.25±0.15 | 93.82±0.07 |
| **Distilbert$_6$-ours** | **79.46±0.08/87.48±0.05** | **93.96±0.01** |

Table 2: Stand-Alone regularizers. SQUAD-V1: The evaluation reported as EM and F1 on the dev set. IMDB: The evaluation reported as accuracy on the test set. Results are the average and the standard deviation of 5 random seeds. Training for 10 epochs.

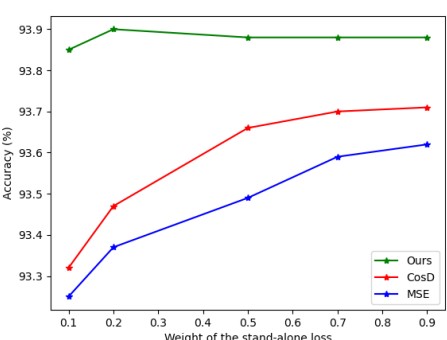

Figure 2: Variation of the model performance on the IMDB dataset in function of the weight of each stand-alone loss in the final loss. For each weight value, the experiments are repeated for 5 seeds and the average is reported.

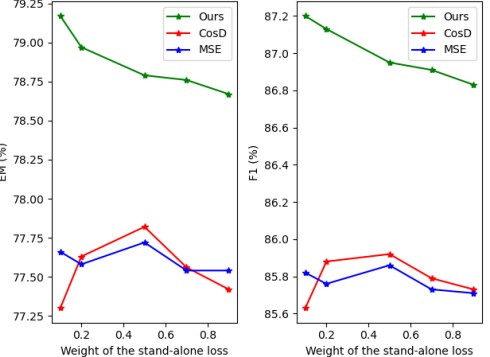

Figure 3: Variation of the model performance on the SQUAD-V1 dataset in function of the weight of each stand-alone loss in the final loss. For each weight value, the experiments are repeated for 5 seeds and the average is reported.

## 4.2 Comparison With Competing Methods

In this evaluation part, we compare the performance of our proposed approach with the competing methods on the commonly used GLUE benchmark dataset (Wang et al., 2018) for knowledge distillation in NLP. For comparison, we fine-tune the student on the tasks without distillation. We report the stand-alone results for MSE and cosine distance. We also generate the results of vanilla KD (Hinton et al., 2015) and PKD (Sun et al., 2019). For the PKD approach, we use a similar setting to (Sun et al., 2019), where the number of epochs is set to 4, the distillation loss ratio is set to 100, the hard loss ratio is set to 0.5, and the temperature

is set to 5. For the rest of the baselines, similar to (Sanh et al., 2019; Zhou et al., 2021), the number of epochs is set to 3. For the vanilla-KD, the temperature is set to 2 (Hinton et al., 2015). All weights in loss functions are fixed to 1 except for the vanilla-KD loss weight is chosen from {0.4, 0.5, 0.6} (Sun et al., 2019) and 0.4 is the best match. In our approach, $\lambda_1$, $\lambda_2$, $\alpha$, and $\beta$ are set to 1, $5.10^{-3}$, 0.5, and $5.10^{-3}$, respectively. After performing a manual tuning, we found out that the aforementioned hyper-parameter settings led to stable results. The number of epochs is set to 3.

The GLUE benchmark dataset is composed of several sub-datasets for different tasks. QQP,

| Approach | MRPC | RTE | CoLA | SST-2 | STS-B | MNLI-m | QNLI | QQP | AVG |
|---|---|---|---|---|---|---|---|---|---|
| BERT$_{12}$(teacher) | 87.86 | 66.79 | 54.84 | 90.02 | 89.30 | 82.85 | 90.70 | 88.04 | 81.30 |
| Distilbert$_6$-FT | 84.48 | 56.17 | 44.28 | 89.54 | 85.40 | 80.41 | 86.71 | 87.79 | 76.85 |
| Distilbert$_6$-CosD | 85.38 | 63.75 | 46.82 | 89.79 | 85.59 | 80.51 | 88.62 | 87.78 | 78.53 |
| Distilbert$_6$-MSE | 86.04 | 62.38 | 47.35 | 89.79 | 85.42 | 80.12 | 88.13 | 87.90 | 78.39 |
| Distilbert$_6$-PKD | 86.06 | 62.24 | 47.28 | 90.05 | **85.77** | 81.62 | 87.16 | **88.33** | 78.56 |
| Distilbert$_6$-KD | 84.56 | 56.53 | 45.86 | 89.63 | 85.50 | 80.51 | 87.71 | 87.60 | 77.23 |
| Distilbert$_6$-**ours** | **86.57** | **63.83** | **50.73** | **90.44** | 85.66 | **82.76** | **89.54** | 88.04 | **79.70** |

Table 3: Results on the GLUE dataset: Evaluation reported on the dev set as the average of 5 random seeds. Training for 3 epochs. AVG is the average performance across the 8 GLUE datasets. All values are in (%).

| Approach | MRPC | RTE | CoLA | SST-2 | STS-B | MNLI-m | QNLI | QQP | AVG |
|---|---|---|---|---|---|---|---|---|---|
| BERT$_{12}$(teacher) | 87.86 | 66.79 | 54.84 | 90.02 | 89.30 | 82.85 | 90.70 | 88.04 | 81.30 |
| Ours | 86.03 | 64.19 | **53.21** | 89.91 | 85.88 | 82.29 | 89.09 | 88.28 | 79.84 |
| Ours-Masked(80%) | 85.79 | 64.55 | 53.16 | **90.18** | 85.94 | **82.41** | **89.44** | 88.32 | **79.97** |
| Ours-Masked(70%) | **86.18** | **64.84** | 52.71 | 89.63 | **85.97** | 82.28 | 89.37 | **88.32** | 79.91 |
| Ours-Masked(Best) | **86.45** | **65.05** | 53.16 | 90.18 | 85.97 | 82.42 | 89.44 | 88.37 | 80.13 |

Table 4: Results on the GLUE dataset: Comparison between Ours and ours-Masked. Evaluation reported on the dev set as the average of 5 random seeds. AVG is the average performance across the 8 GLUE datasets. All values are in (%). Between parenthesises is the percentage (%) of the learned sub-tasks in each iteration. Training for 5 epochs.

MRPC, and STS-B are for the paraphrase similarity matching task. SST-2 is for the sentiment classification task. MNLI-m (matched version of MNLI), QNLI, and RTE are for the natural language inference task. CoLA is for the linguistics acceptability task. For MRPC and QQP we report the combined score from F1 and accuracy. For STS-B we report the combined score from Pearson correlation and Spearman correlation. For CoLA we report the Matthew's correlation. For the rest of the tasks, accuracy is the metric.

As shown in Table 3, our approach outperforms KD, which stands for vanilla-KD (Hinton et al., 2015), and the PKD (Sun et al., 2019) baselines on most of the GLUE tasks. It is worth mentioning that for the SST-2 and QQP datasets, the resulting student model using our approach and the one using the PKD approach could exceed the performance of the teacher. This is commonly observed in (Stanton et al., 2021; Furlanello et al., 2018). Furthermore, it consistently outperforms the MSE and the cosine distance distillation objective functions, which are applied between the last hidden layer of the teacher and the last hidden layer of the student, on all the GLUE tasks. As shown in the last column of Table 3, on all the GLUE tasks, our One-to-One Mapping approach achieves the highest average performance, i.e., 79.70%, with up to 1.14% compared to the results of the other approaches. Our proposed approach helped to effectively reduce the

gap between the teacher average performance, i.e., 81.30% and the student average performance, i.e., 79.70% on all the GLUE tasks.

## 5 Ablation Study

See results in Table 5 and as shown in the table both terms of the loss are important and their combination achieves the best results across most of the datasets. Thus, the two components of the loss have a compound effect. $L_{CE}$ is the hard loss. $l$ is the first term in the $L_{KD}$ loss. $R$ is the second term in the $L_{KD}$ loss (the repel term).

## 6 Analalyis

### 6.1 Layer Distillation: Our loss as an alternative for MSE and CosD

Rather than looking at our approach as a competing method to the KD state-of-the-art approaches, it is proposed to replace any MSE or cosine distance-based feature distillation methods between two hidden layers included in any knowledge distillation approach. To show that our approach can outperform MSE and cosine distance, applied globally, in the existing pipelines, we experiment with SRRL (Yang et al., 2020), one of the latest KD approaches. SRRL KD loss has two components: the feature loss, which is an MSE between the last layer's features of the teacher and the last layer's features of the student,

Table 5: Ablation Study (Ours). All values in (%). All the results are the average over 5 trials

| approach | MRPC | RTE | CoLA | SST-2 | STS-B | QNLI | MNLI-m | QQP | AVG |
|---|---|---|---|---|---|---|---|---|---|
| $L_{CE} + l$ | **86.85** | 63.61 | 48.24 | **90.87** | 85.47 | **89.63** | 81.85 | 87.60 | 79.26 |
| $L_{CE} + R$ | 75.85 | 53.79 | 34.71 | 85.25 | 85.25 | 85.66 | 74.43 | 87.69 | 68.49 |
| $L_{CE} + R + l$ | 86.57 | **63.83** | **50.73** | 90.44 | **85.66** | 89.54 | **82.76** | **88.04** | **79.70** |

and the logit loss. In this experiment, we replace the feature KD loss in SRRL with our newly introduced correlation loss function. As shown in the Table 6, our feature distillation approach replacing MSE loss yields superior performance for SRRL on the two GLUE datasets.

Table 6: SRRL vs SRRL-Ours

| approach | QNLI(%) | RTE(%) |
|---|---|---|
| SRRL | 88.04 | 63.68 |
| SRRL-Ours | 88.68 | 64.10 |

## 6.2 Model Efficiency

Although our distillation objective is applied as a stand-alone between the last hidden layer of the student and the last hidden layer of the teacher, it could outperform PKD. Note that PKD (Sun et al., 2019) distills different hidden-state features plus the logits. However, our approach distills only the last hidden feature of the teacher to the student. Our approach has a lower computational, as shown in Table 7, our approach is trained in less time. In this experiment, we report the training time of PKD and Ours on 3 different datasets. Training was done on a single A100 GPU with a fixed batch-size of 16.

## 6.3 Sensitivity to the Batch Size

Our proposed loss function depends on the batch-size. As shown in Table 8, in all the datasets, the performance of our approach is consistent and stable across all the batch-size values. This illustrates the robustness of our proposed method and shows that it can work well even with small batch-size values. However, it is clear when the batch-size increases, the results marginally increase on most of the datasets. This is logical because by increasing the number of samples, the correlation value become more accurate.

## 7 Masked One-to-One Mapping

In our One-to-One Mapping feature distillation approach, the goal for the student is to mimic the last hidden layer representation of the teacher. To achieve this, we break down the global feature distillation task into $N$ local sub-tasks. To further enhance the student performance, we propose an augmented variant of our approach called Masked One-to-One Mapping. Instead of learning all $N$ sub-tasks simultaneously, it is more effective for the student to focus on a subset of the total sub-tasks at a time. This approach enables the student to effectively absorb and manage the incoming flow of knowledge from the teacher.

As shown in Figure 4, during each iteration (optimization step), we randomly mask a subset of units in the last hidden layer of the teacher and in the last hidden layer of the student. Afterwards, we will have $m$ (where $m < N$) neurons, i.e., $m$ feature distillation sub-tasks left for the student to learn. We accomplish this by applying a same binary mask $M$ to the last hidden layer features representation of both the student and the teacher simultaneously.

Formally, given $Y_s \in \mathbf{R}^{N \times n}$ and $Y_t \in \mathbf{R}^{N \times n}$, the features representation of the student and the teacher, respectively. $N$ is the number of neurons in the last hidden layers of the teacher and the student. $n$ is the batch size. To generate a random binary mask $M$, we create a vector of size $N$ containing independent Binomial random variables. Each variable in this vector takes the value 1 with a probability $p$ and 0 otherwise. We multiply element-wise this binary mask $M$ by each column of $Y_t$ and $Y_s$ to generate the masked features representation $Y'_t$ of the teacher and $Y'_s$ of the student, respectively:

$$M \sim Bernoulli(p)$$
$$Y'_t = [Y_t^1 \odot M, Y_t^2 \odot M, Y_t^3 \odot M, ..., Y_t^n \odot M]$$
$$Y'_s = [Y_s^1 \odot M, Y_s^2 \odot M, Y_s^3 \odot M, ..., Y_s^n \odot M]$$

Where $Y_t^1$ represents the first column (the feature representation of the first sample in the input batch $X$) of the features representation $Y_t$ of the teacher. $\odot$ denotes the element wise multiplication. After removing the masked rows, i.e., rows with all elements zeros, from $Y'_t$ and $Y'_s$, we obtain the new features representation of the teacher $Y'_t \in \mathbf{R}^{m \times n}$ and the student $Y'_s \in \mathbf{R}^{m \times n}$, where $m < N$.

In the next steps, we apply the same One-to-One Mapping approach described in section 3. How-

Table 7: Training time of PKD and Ours. The results are in hours (H)

| approach | MNLI-m (H) | QNLI(H) | SST-2(H) |
|---|---|---|---|
| PKD | 3.00 | 0.73 | 0.53 |
| Ours | **2.13** | **0.63** | **0.47** |

Table 8: The performance of our approach (Ours) across different batch-size values. Average over 5 trials.

| Batch-size | MNLI-m (%) | QNLI (%) | SST-2 (%) | RTE (%) |
|---|---|---|---|---|
| 4 | 82.41 | 89.43 | 90.18 | 63.24 |
| 8 | 82.54 | 89.53 | 90.44 | 65.85 |
| 16 | 82.76 | 89.54 | 90.44 | 63.83 |

ever, instead of having $N$ specialized sub-teachers and $N$ focused sub-students, in our Masked One-to-One Mapping, we have $m$ specialized sub-teachers and $m$ focused sub-students. Furthermore, we have $m$ sub-tasks where each focused sub-student attempts to mimic the feature representation of the corresponding specialized sub-teacher while disregarding the others. To achieve this, we utilize the modified training objective function 6.

$$L'_{KD} = \lambda_1 \sum_i^m (1 - C_{ii})^2 + \lambda_2 \sum_i^m \sum_{j \neq i}^m C_{ij}^2 \quad (9)$$

We conduct experiments on 8 GLUE datasets using the same setup as described in subsection 4.2. However, in these new experiments, we extend the training time to 5 epochs because as shown in Tables 3 and 4, the average performance on the GLUE tasks of our student model is higher when trained for 5 epochs compared to when trained for 3 epochs. We introduce $p$ as the percentage (%) of sub-tasks to be learned at a given optimization step. We conduct experiments for all values of $p \in \{10, 20, 30, 40, 50, 70, 90\}$. However, we report the results of the two best setups, in function of $p$, in terms of the average performance across all 8 GLUE datasets. Additionally, we provide the best performance achieved for each individual dataset. For each dataset, the experiment is repeated for 5 seeds and the average is reported.

As illustrated in Table 4, the average performance on the GLUE datasets for the Masked One-to-One Mapping approach, referred to as Ours-Masked, is higher than the value obtained by the One-to-One Mapping, referred to as Ours, for the two $p$ values 80% and 70%. When considering the best performance across all $p$ values for each individual dataset, denoted as Ours-Masked(Best), the gap between the teacher performance (81.30%) and the student performance (80.13%) is effectively reduced.

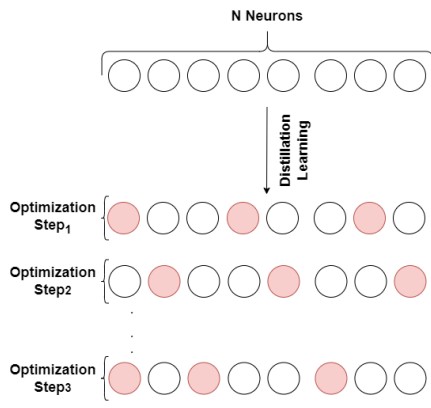

Figure 4: Masked One-to-One Mapping: In each optimization step, the neurons that are marked by the red color will be masked.

# 8 Conclusion and Future Work

In this paper, we reformulated the global feature distillation problem into $N$ local sub-problems. We proposed a one-to-one mapping between each neuron in the last hidden layer of the teacher, i.e., specialized sub-teacher, and each neuron in the last hidden layer of the student. i.e., focused sub-student. To achieve this goal, we proposed a local correlation-based loss across batches between each specialized sub-teacher output and its corresponding focused sub-student output. Our approach only requires the teacher and the student to have the same last hidden layer size. Several experiments proved the effectiveness and consistency of our method. Our approach can be added to any KD method in NLP or vision. Moreover, we proposed an augmented variant of our One-to-One Mapping approach called Masked One-to-One Mapping. Instead of learning all the sub-tasks simultaneously, we make the student learn a subset of the tasks at a time. Future work includes exploring the same distillation process with several intermediate layers and experimenting with our approach beyond the NLP tasks.

## Limitations

In this work, the proposed approach requires the student and the teacher to have the same last hidden layer size, which might be a limitation. One possible solution could be to investigate the impact of adding a projector after the last hidden layer of the teacher to make it match the last hidden layer size of the student.

## Ethics Statement

This work is mainly about task-specific knowledge distillation in NLP. It proposes a new viewpoint on how to study the feature distillation task. This new approach does not pose any ethical issues as the utilized datasets are publicly available and have already been used in previous research.

## Acknowledgment

The project on which this report is based was funded by the German Federal Ministry of Education and Research (BMBF) under the funding code 01|S20049. The author is responsible for the content of this publication.

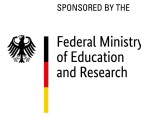

SPONSORED BY THE

Federal Ministry of Education and Research

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
