# OpenReview forum: "Learn From One Specialized Sub-Teacher: One-to-One Mapping for Feature-Based Knowledge Distillation"
_EMNLP/2023/Conference — EMNLP 2023 Findings_

### Official Review · Reviewer_BBGK · 2023-08-02

**Soundness:** 3

**Excitement:**

2: Mediocre: This paper makes marginal contributions (vs non-contemporaneous work), so I would rather not see it in the conference.

**Missing References:**

The paper is missing citations/links to the datasets used in the experiments (SQUAD, IMDB).

**Paper Topic And Main Contributions:**

This paper proposes a feature distillation loss function that aligns the student model’s last hidden layer representation with the teacher’s neuron by neuron. This loss function is different from the existing MSE or cosine distance loss functions to align the hidden layer representations because MSE or cosine distance penalizes different representations per sample while the proposed one-to-one mapping in this paper encourages similar representations per feature (along each dimension across samples in the same batch).

In addition, the authors propose a variation of this one-to-one mapping, where only a subset of the neurons in the last hidden layers are optimized for the per-feature distillation loss function. The authors claim this variation introduces additional gains.

Overall, the empirical results in the paper show that the idea is promising.

**Questions For The Authors:**

Please see my questions in the section "Reasons to Reject."

**Reasons To Accept:**

- The paper clearly illustrates the proposed idea with helpful visualizations and precise mathematical formulas.

- The empirical results in the paper show that the proposed loss function for knowledge distillation leads to enhanced student model performance compared to existing methods. Results are repeated runs and thus are convincing in terms of statistical significance.

**Reasons To Reject:**

1. Some claims are not well justified.

    (a) Line 194 states the proposed loss function is more stable, but I do not find theoretical explanations or empirical evidence that supports this statement. In what aspects do we consider the proposed method more stable – less sensitive to unit permutation, robust to hyper-parameter choice, or something else?

    (b) Line 295 claims the proposed method can be done with a low computation cost. However, there are no relevant statistics such as GPU hours to support the claim.

    (c) Why does a higher EM, F1, or accuracy prove the proposed approach “facilitates the convergence of the student’s representation to the teacher’s representation” (line 348)? Why can’t it be the case that the student model represents the features vastly differently from the teacher model but has learned a better representation for the task? Although the claim is making sense, one would need to specifically look at the hidden representations in order to show this convergence.

    (d) Line 462 states that masked one-to-one mapping “enables the student to effectively absorb and manage the incoming flow of knowledge from the teacher.” However, the empirical results in Table 3 (where no masking is applied) contain marginally higher accuracy for several tasks (e.g., MRPC, SST-2, etc.). This claim sounds untenable.

2. The paper does not provide sufficient details on the datasets used in the experiments. The authors should not assume that a reader is familiar with what task SQUAD-V1 or IMDB is addressing and what evaluation metrics are for the dataset.

3. The analysis of the proposed loss function is not thorough enough.

    (a) Does each component of the loss function contribute to the effectiveness of the proposed method? Why is it necessary to have the second component with hyperparameter $\lambda_2$ that aims to “repel all knowledge coming from the other teachers”? This could be potentially counter-productive if there exist features that are highly correlated with each other. In fact, I see that $\lambda_2 = 5 \times 10^{-3}$ is much smaller than $\lambda_1=1$ in the experiment section (line 403).

    (b) The cross-correlation contains a variable $n$, which is the batch size. Does the proposed method work equally well regardless of the batch size, or does it not work well if the $n$ is small?

**Reproducibility:**

4: Could mostly reproduce the results, but there may be some variation because of sample variance or minor variations in their interpretation of the protocol or method.

**Reviewer Confidence:**

4: Quite sure. I tried to check the important points carefully. It's unlikely, though conceivable, that I missed something that should affect my ratings.

**Typos Grammar Style And Presentation Improvements:**

- The cosine distance formula contains typos in the subscripts.

- The loss functions in Fig. 1 are missing the squaring operation around $(1-C)$.

---

> ### Author Rebuttal · Authors · 2023-08-28
>
> Thank you for your time and your interesting feedback.  We include clarifications to individual points below. If there are any remaining questions, please let us know.
>
> >(a) Line 194 states the proposed loss function is more stable, but I do not find theoretical explanations or empirical evidence that supports this statement. In what aspects do we consider the proposed method more stable – less sensitive to unit permutation, robust to hyper-parameter choice, or something else?
>
> Stability in this context is used in a sense that the results are stable which is reflected by a low value of standard deviation over the 5 random seeds. As you can see in Table 1 and Table 2, our results gave lower standard deviation with respect to different random seeds compared to the other approaches. We will make this clear in the final version of the paper.
>
> >(b) Line 295 claims the proposed method can be done with a low computation cost. However, there are no relevant statistics such as GPU hours to support the claim.
>
> By low computation, we meant lower compared to previous approach, more specifically to PKD [1]. Note that, unlike PKD [1], which distills different hidden-state features plus the logits, our approach distills only the last hidden feature of the teacher to the student. Moreover, we can now confirm this with quantitative results by reporting the training time of each approach on 3 different datasets. We chose 3 big-size datasets from GLUE in order to have a fair comparison. Training was done on a single A100 GPU with a fixed batch-size of 16. As show in Table below, our approach is trained in less time. We will include these results in the final version of the paper. Thank you for your suggestion.
>
>
> | approach | MNLI-m (H)  | QNLI (H)       | SST-2 (H)      |   |
> |----------|-------------|---------------|---------------|---|
> | PKD      | 3.00        | 0.73          | 0.53          |
> | Ours     | **2.13** | **0.63** |**0.47** |
>
> The results are in hours (H).
>
> >(c) Why does a higher EM, F1, or accuracy prove the proposed approach “facilitates the convergence of the student’s representation to the teacher’s representation” (line 348)? Why can’t it be the case that the student model represents the features vastly differently from the teacher model but has learned a better representation for the task? Although the claim is making sense, one would need to specifically look at the hidden representations in order to show this convergence.
>
> Our approach is formulated directly as maximizing the correlation between each unit in the student and its corresponding unit in the teacher. Thus, it is safe to assume that after training our model, the student's features representation is highly correlated to the teacher's features representation or in other words very similar to it. But we agree with the reviewer's point of view, this claim can be softened in the final version of the paper and we can replace it with "yields better representation of the student features"
>
> >(d) Line 462 states that masked one-to-one mapping “enables the student to effectively absorb and manage the incoming flow of knowledge from the teacher.” However, the empirical results in Table 3 (where no masking is applied) contain marginally higher accuracy for several tasks (e.g., MRPC, SST-2, etc.). This claim sounds untenable.
>
> This claim is based on the fact that Ours-Masked outperforms ours in average across all the datasets. However, the reviewer has a point, on some datasets out of the 8, ours outperforms Ours-Masked. Interestingly this happens mostly when the dataset is of small size. We will include this analysis in the final version of the paper.
>
> >**Q2:** The paper does not provide sufficient details on the datasets used in the experiments. The authors should not assume that a reader is familiar with what task SQUAD-V1 or IMDB is addressing and what evaluation metrics are for the dataset.
>
> Thank you for pointing this out.
>
> *Stanford Question Answering Dataset (SQuAD)* [2]
> is a reading comprehension dataset, consisting of questions posed by crowd workers on a set of Wikipedia articles, where the answer to every question is a segment of text, or span, from the corresponding reading passage. \
> *Exact Match (EM)* metric [2]: measures the percentage of predictions that match any one of the ground truth answers exactly.\
> *F1 score metric* [2]: measures the average overlap between the prediction and ground truth answer. \
> *Internet Movie Database dataset (IMDB)* [3]  is a sentiment analysis dataset consisting of 50,000 movie reviews labeled as positive or negative. \
> We will include this description in the final version of the paper.
>
> >(a) Does each component of the loss function contribute to the effectiveness of the proposed method? ...
>
> Our approach is composed of two parts. The first term makes each sub-student to focus with the corresponding sub-teacher. The second term helps to make the task easy for each sub-student. We need to remember that the student has lower number of parameters compared to the teacher. To this end, we force each sub-student to ignore the redundant knowledge learnt by the other sub-teachers and to focus only on the distinctive information that it can get from its corresponding sub-teacher. As shown by the ablation study we add now, this second term helps to boost the results of most of the datasets.
>
> | approach     | MRPC         | RTE            | CoLA           | SST-2          | STS-B          | QNLI           | MNLI-m         | QQP            | AvG            |
> |--------------|--------------|----------------|----------------|----------------|----------------|----------------|----------------|----------------|----------------|
> | $L_{CE}+l$   | **86.85** | 63.61          | 48.24          | **90.87** | 85.47          | **89.63** | 81.85          | 87.60          | 79.26          |
> | $L_{CE}+R$   | 75.85        | 53.79          | 00.00          | 85.25          | 85.25          | 85.66          | 74.43          | 87.69          | 68.49          |
> | $L_{CE}+R+l$ | 86.57        |**63.83** | **50.73** | 90.44          | **85.66** | 89.54          |**82.76**|**88.04**| **79.70** |
>
> All the values in the table are in (\%). Reported results are the average over 5 random seeds.\
> $L_{CE}$: is the hard loss.\
> $l$: is the first term in the $L_{KD}$ loss.\
> $R$: is the second term in the $L_{KD}$ loss (the repel term).
>
> >(b) The cross-correlation contains a variable, which is the batch size. Does the proposed method work equally well regardless of the batch size, or does it not work well if the n is small?
>
> (b) Indeed, our proposed loss function depends on the batch-size. In our experiments, we found that the results are not very sensitive to the batch size. To show that, now, we include the results in Table below with different batch-size values. The chosen datasets are with different sizes (big, medium, and small). As shown in Table below, in all the datasets, the performance of our approach is consistent and stable across all the batch-size values.  This illustrates the robustness of our proposed method and  shows that it can work well even with small batch-size values. However, we note when the batch-size increases, the results marginally increase on most of the datasets. This is reasonable because by increasing the number of samples, the correlation value become more accurate. We will include these results in the final version of the paper.
>
> | Batch-size | MNLI-m (\%) | QNLI (\%) | SST-2 (\%) | RTE (\%) |
> |------------|-------------|-----------|------------|----------|
> | 4          | 82.41       | 89.43     | 90.18      | 63.24    |
> | 8          | 82.54       | 89.53     | 90.44      | 65.85    |
> | 16         | 82.76       | 89.54     | 90.44      | 63.83    |
>
>
>
> >(a) The cosine distance formula contains typos in the subscripts.(b)The loss functions in Fig. 1 are missing the squaring operation around.
>
> Thank you for mentioning this, we will fix them in the final version of the paper.
>
> [1] Patient Knowledge Distillation for BERT Model Compression, S. Sun and Yu Cheng and Zhe Gan and Jingjing Liu, EMNLP2019.\
> [2] SQuAD: 100,000+ Questions for Machine Comprehension of Text. Rajpurkar, Pranav  and Zhang, Jian  and Lopyrev, Konstantin  and Liang, Percy. EMNLP2016.\
> [3] Learning Word Vectors for Sentiment Analysis. Maas, Andrew L.  and Daly, Raymond E.  and Pham, Peter T.  and Huang, Dan  and Ng, Andrew Y.  and Potts, Christopher. ACL2011.

---

### Official Review · Reviewer_tgjk · 2023-08-04

**Soundness:** 3

**Excitement:**

3: Ambivalent: It has merits (e.g., it reports state-of-the-art results, the idea is nice), but there are key weaknesses (e.g., it describes incremental work), and it can significantly benefit from another round of revision. However, I won't object to accepting it if my co-reviewers champion it.

**Missing References:**

None

**Paper Topic And Main Contributions:**

The paper introduces a novel approach to knowledge distillation based on features by aligning every output of neurons from teacher and student. Instead of the usually used MSE, the paper proposes to maximize the correlation between mapped outputs and reduce that from unmapped ones.

**Questions For The Authors:**

See above.

**Reasons To Accept:**

The proposed distillation method is novel. And the proposed idea is easy to follow. Besides, compared to the baseline, the experiments show that the proposed method have promising results.

The implementation is also straight, which may be practical in real scenarios.

The illustration about other metric's sensitive to unit permutation is intuitive.


**Reasons To Reject:**

1. More advanced knowledge distillation methods should be compared. For example, Contrastive Representation Distillation, Decoupled KD, etc.
2. I personally find the novelty in this paper rely on the design of selecting the correlation as the metric, meanwhile reduce the correlation between different neurons. This has a similar insight with dropout by increasing the independence of every unit. I think the intuition behind these strategies can be more explored. I know in the relative work it has been discussed somehow, however, I think it can be more detailed, by showing why it is important to reduce the correlation between different neurons, and why correlation as metric is superior. Plus, the paper writes "In a high dimensional space, which is the case in a neural network, the data tend to be sparse and all the data points become uniformly distant from each other", which is not the real case in deep neural network as far as I know, especially Transformer-based networks.

**Reproducibility:**

4: Could mostly reproduce the results, but there may be some variation because of sample variance or minor variations in their interpretation of the protocol or method.

**Reviewer Confidence:**

4: Quite sure. I tried to check the important points carefully. It's unlikely, though conceivable, that I missed something that should affect my ratings.

**Typos Grammar Style And Presentation Improvements:**

Please check if the equation at line 184 is mistaken.

---

> ### Author Rebuttal · Authors · 2023-08-28
>
> Thank you for your time and your interesting feedback.  We include clarifications to individual points below. If there are any remaining questions, please let us know.
>
> >**Q1:** More advanced knowledge distillation methods should be compared. For example, Contrastive Representation Distillation, Decoupled KD, etc.
>
> The main contributions of our work, first, is identifying the drawbacks of the existing objective functions (MSE and cosine distance) used to perform the feature distillation task between two transformer layers. Second, breaking down the global layer distillation task into multiple local sub-tasks and introducing a correlation-based loss function to alleviate the limitation of previous loss functions (MSE and cosine distance-based). Thus, rather than looking at our approach as a competing method to the KD state-of-the-art approaches, it is proposed to replace any MSE or cosine distance-based feature distillation methods between two hidden layers included in any knowledge distillation approach. \
> To show that our approach can outperform MSE and cosine distance, applied globally, in the existing pipelines, we experiment now the with SRRL[2], one of the latest KD approaches.
> SRRL KD loss has 2 components: the feature loss, which is an MSE between the last layer's feature of the teacher and the last layer's feature of the student, and the logit loss, which is an MSE between the teacher logits and the student feature after being passed through the teacher classifier.\
> In the experiment below, we replace the feature KD loss in SRRL with our newly introduced correlation loss function.
> As shown in the Table below, our feature distillation approach replacing global MSE yields superior performance for SRRL.
>
> | approach  | QNLI(\%) | RTE(\%) |
> |-----------|----------|---------|
> | SRRL      | 88.04    | 63.68   |   |
> | SRRL-Ours | **88.68**    | **64.10**   |   |
>
> **SRRL-Ours: SRRL with its the feature distillation loss replaced by our proposed loss.**\
> We will include these results in the final version of the paper.
>
> To sum this up, our approach can be incorporated in any KD approach to replace MSE/cosine distance, which are usually used [1-3] in feature distillation, with our correlation based loss. In this example, we applied it to SRRL. However, this can be incorporated in [1,3]. In this sense, our method is not competing with KD approaches. However, it is a new alternative for using MSE and cosine distance as global metrics in feature distillation.
>
> >(a) I personally find the novelty in this paper rely on the design of selecting the correlation as the metric, meanwhile reduce the correlation between different neurons. This has a similar insight with dropout by increasing the independence of every unit. I think the intuition behind these strategies can be more explored. I know in the relative work it has been discussed somehow, however, I think it can be more detailed, by showing why it is important to reduce the correlation between different neurons.
>
> The reviewer makes an interesting point. The connection between our approach and dropout is interesting. However, from our perspective, our proposed approach is different compared to the dropout mechanism. It is true that dropout can reduce  the correlation between the units within the same layer. But, in our approach we reduce the correlation between student units and teacher units in order to facilitate the sub-task for the sub-student. In other words, the correlation is computed between the corresponding units from two different models.
>
> >(b) Why correlation as metric is superior.
>
> As mentioned in the introduction and the related work, MSE and cosine distance suffer from various limitations such as sensitivity to scale and sensitivity to permutation. Kindly refer to Line 060 and Line 184  to see the drawbacks of using MSE and cosine distance.
>
> >(c) Plus, the paper writes "In a high dimensional space, which is the case in a neural network, the data tend to be sparse and all the data points become uniformly distant from each other", which is not the real case in deep neural network as far as I know, especially Transformer-based networks.
>
> There seems to be a misunderstanding of our claim, what we wanted to say is that the embeddings (features representation) provided by BERT-like models are high-dimensional and sparse tensors in nature, see [4]. Thus, using MSE or cosine distance to measure the distance between these sparse representations is not optimal as explained and advocated in [5]. So, splitting the task into sub-tasks and using one-to-one metric alleviates this problem and breaks the curse of dimensionality. We will make this more clear in the final version of the paper.
>
> >Typos Grammar Style And Presentation Improvements:Please check if the equation at line 184 is mistaken.
>
> Thank you for pointing out this. We will fix it in the final version of the paper.
>
> [1] Patient Knowledge Distillation for BERT Model Compression, S. Sun and Yu Cheng and Zhe Gan and Jingjing Liu, EMNLP2019.\
> [2] Knowledge distillation via softmax regression representation learning, Jing Yang, Brais Martinez, Adrian Bulat, Georgios Tzimiropoulos. ICLR2021.\
> [3] TinyBERT: Distilling BERT for Natural Language Understanding. Xiaoqi Jiao, Yichun Yin, Lifeng Shang, Xin Jiang, Xiao Chen, Linlin Li, Fang Wang, Qun Liu. EMNLP Findings 2020.\
> [4] THE LAZY NEURON PHENOMENON: ON EMERGENCE OF ACTIVATION SPARSITY IN TRANSFORMERS ,Zonglin Li , Chong You , Srinadh Bhojanapalli, Daliang Li, Ankit Singh Rawat, Sashank J. Reddi, Ke Ye, Felix Chern, Felix Yu, Ruiqi Guo, and Sanjiv Kumar. ICLR2023.\
> [5] On the Surprising Behavior of Distance Metrics in High Dimensional Space, Charu C. Aggarwal1, Alexander Hinneburg, and Daniel A. Keim2. ICDT 2001.

---

### Official Review · Reviewer_pptK · 2023-08-09

**Soundness:** 2

**Excitement:**

2: Mediocre: This paper makes marginal contributions (vs non-contemporaneous work), so I would rather not see it in the conference.

**Paper Topic And Main Contributions:**

This paper is about knowledge distillation, which is a technique for compressing large and complex neural networks into smaller and more efficient ones. The paper addresses the problem of how to transfer the knowledge of the teacher network, which is the large model, to the student network, which is the small model, by using the intermediate feature representations of both networks. The main contributions of the paper are:

1. It proposes a novel approach called One-to-One Mapping, where each neuron in the last hidden layer of the teacher network is considered as a specialized sub-teacher, and each neuron in the last hidden layer of the student network is considered as a focused sub-student. The paper argues that this approach can better align the feature representations of the teacher and the student than using global metrics such as mean square error or cosine distance.

2. It introduces a local correlation-based objective function to achieve the One-to-One Mapping distillation, where each focused sub-student tries to maximize the correlation with its corresponding specialized sub-teacher and minimize the correlation with other sub-teachers. The paper claims that this objective function is more stable and accurate than using mean square error or cosine distance.

3. It conducts extensive experiments on several natural language processing tasks and datasets, such as SQUAD-V1, IMDB, and GLUE, and shows that its proposed approach outperforms the state-of-the-art methods by maintaining higher performance on most benchmark datasets. The paper also presents a randomized variant of its approach, called Masked One-to-One Mapping, where the student learns a subset of sub-tasks at each optimization step. The paper reports that this variant yields superior results than learning all sub-tasks simultaneously.


**Questions For The Authors:**

1. How do you ensure that the one-to-one mapping between the teacher and student neurons is consistent across different batches and epochs? Do you use any regularization or alignment techniques to prevent the mapping from changing over time?
2. How do you handle the case where the student and teacher have different last hidden layer sizes? Do you use any projection or dimensionality reduction methods to make them compatible?

**Reasons To Accept:**

1. The paper proposes a novel and effective approach for feature-based knowledge distillation, which is a technique for compressing large and complex neural networks into smaller and more efficient ones.
2. The paper introduces a new perspective on feature distillation, where each neuron in the last hidden layer of the teacher network is considered as a specialized sub-teacher, and each neuron in the last hidden layer of the student network is considered as a focused sub-student. The paper argues that this perspective can better align the feature representations of the teacher and the student than using global metrics such as mean square error or cosine distance.
3. The paper conducts extensive experiments on several natural language processing tasks and datasets, such as SQUAD-V1, IMDB, and GLUE, and shows that its proposed approach outperforms the state-of-the-art methods by maintaining higher performance on most benchmark datasets. The paper also presents a randomized variant of its approach, called Masked One-to-One Mapping, where the student learns a subset of sub-tasks at each optimization step. The paper reports that this variant yields superior results than learning all sub-tasks simultaneously.


**Reasons To Reject:**

1. The paper does not provide a clear theoretical justification or analysis for why the proposed One-to-One Mapping approach works better than the global metrics such as MSE or cosine distance. It only relies on empirical results to support its claims, which might not be convincing enough for some reviewers or readers.
2.The paper does not compare its approach with other recent methods for feature-based knowledge distillation, it needs to be supplemented.
3. The paper does not conduct ablation studies to investigate the impact of different components or hyperparameters of the proposed approach. It also does not provide any error analysis or qualitative examples to illustrate the strengths and limitations of the proposed approach in different scenarios or tasks.



**Reproducibility:**

2: Would be hard pressed to reproduce the results. The contribution depends on data that are simply not available outside the author's institution or consortium; not enough details are provided.

**Reviewer Confidence:**

3: Pretty sure, but there's a chance I missed something. Although I have a good feel for this area in general, I did not carefully check the paper's details, e.g., the math, experimental design, or novelty.

---

> ### Author Rebuttal · Authors · 2023-08-28
>
> Thank you for your time and your interesting feedback.  We include clarifications to individual points below. If there are any remaining questions, please let us know.
>
> >**Q1**:How do you ensure that the one-to-one mapping between the teacher and student neurons is consistent across different batches and epochs? Do you use any regularization or alignment techniques to prevent the mapping from changing over time?
>
> The mapping between each sub-student and each sub-teacher is fixed, i.e. based on an index, and thus remains consistent over batches and epochs (see Equations and Figure 1). For example, the sub-teacher$_1$ (the first neuron in the teacher's last hidden layer) is mapped to the sub-student$_1$ with the same index (the first neuron in the student's last hidden layer), the sub-teacher$_2$ (the second neuron in the teacher's last hidden layer) is mapped to the sub-student$_2$ with the same index (the second neuron in the student's last hidden layer), etc. So, this mapping is fixed over epochs and batches.
>
> >**Q2:** How do you handle the case where the student and teacher have different last hidden layer sizes? Do you use any projection or dimensionality reduction methods to make them compatible?
>
> Kindly refer to Line 079 of the paper and to the conclusion section. In our work, we assume that the teacher and student's last hidden states have the same dimension, as in [1]. We also mentioned that future work will include exploring the addition of a projector to address this limitation.
>
> >**Q3:** The paper does not provide a clear theoretical justification or analysis for why the proposed One-to-One Mapping approach works better than the global metrics such as MSE or cosine distance.
>
> We have discussed the drawbacks using MSE (e.g. sensitive to scale Lines 065/184 ) and the cosine distance (e.g. sensitive to permutation Line 184) objective functions in the introduction (Line 060)  and the related work (Line 205), which we think is the relevant theoretical justification for our approach. We also linked different references and examples to support our claims.
>
> >**Q4:** The paper does not compare its approach with other recent methods for feature-based knowledge distillation, it needs to be supplemented.
>
> The main contributions of our work, first, is identifying the drawbacks of the existing objective functions (MSE and cosine distance) used to perform the feature distillation task between two transformer layers. Second, breaking down the global layer distillation task into multiple local sub-tasks and introducing a correlation-based loss function to alleviate the limitation of previous loss functions (MSE and cosine distance-based). Thus, rather than looking at our approach as a competing method to the KD state-of-the-art approaches, it is proposed to replace any MSE or cosine distance-based feature distillation methods between two hidden layers included in any knowledge distillation approach. \
> To show that our approach can outperform MSE and cosine distance, applied globally, in the existing pipelines, we experiment now with SRRL[2], one of the latest KD approaches.
> SRRL KD loss has 2 components: the feature loss, which is an MSE between the last layer's features of the teacher and the last layer's features of the student, and the logit loss, which is an MSE between the teacher logits and the student feature after being passed through the teacher classifier.\
> In the experiment below, we replace the feature KD loss in SRRL with our newly introduced correlation loss function.
> As shown in the Table below, our feature distillation approach replacing  MSE loss yields superior performance for SRRL on two GLUE datasets.
>
> | approach  | QNLI(\%) | RTE(\%) |
> |-----------|----------|---------|
> | SRRL      | 88.04    | 63.68   |   |
> | SRRL-Ours | 88.68    | 64.10   |   |
>
> **SRRL-Ours: SRRL with its feature distillation loss replaced by our proposed loss.**\
> We will include these results in the final version of the paper.\
> To sum this up, our approach can be incorporated in any KD approach to replace MSE/cosine distance, which are usually used in feature distillation [1-3] , with our correlation based loss. In this example, we applied it to SRRL. However, this can be incorporated in [1,3]. In this sense, our method is not competing with KD approaches. However, it is a new alternative for using MSE and cosine distance as global metrics in feature distillation.
>
> >**Q5:** The paper does not conduct ablation studies to investigate the impact of different components or hyperparameters of the proposed approach.
>
> We thank the reviewer for the suggestion. Now, we add an ablation study to investigate the impact of different terms of our algorithm . We report the results in Table below. As can be seen, both terms of the loss are important and their combination achieves the best results across most of the datasets. Thus, the two components of the loss have a compound effect. We will include these results and discussion in the final version of the paper.
>
> | approach     | MRPC         | RTE            | CoLA           | SST-2          | STS-B          | QNLI           | MNLI-m         | QQP            | AvG            |
> |--------------|--------------|----------------|----------------|----------------|----------------|----------------|----------------|----------------|----------------|
> | $L_{CE}+l$   | **86.85** | 63.61          | 48.24          | **90.87** | 85.47          | **89.63** | 81.85          | 87.60          | 79.26          |
> | $L_{CE}+R$   | 75.85        | 53.79          | 34.71          | 85.25          | 85.25          | 85.66          | 74.43          | 87.69          | 68.49          |
> | $L_{CE}+R+l$ | 86.57        | **63.83**| **50.73** | 90.44          |**85.66**| 89.54          |**82.76** | **88.04** | **79.70**|
>
> All the values in the table are in (\%). Reported results are the average over 5 random seeds.\
> $L_{CE}$: is the hard loss\
> $l$: is the first term in the $L_{KD}$ loss.\
> $R$: is the second term in the $L_{KD}$ loss (the repel term).
>
>
> >**Q6:** It also does not provide any error analysis or qualitative examples to illustrate the strengths and limitations of the proposed approach in different scenarios or tasks.
>
> In terms of error analysis, we followed the standard approach found in related work, which usually does not contain qualitative examples. We evaluated our approach on 8 GLUE benchmark datasets. The GLUE dataset, which consists of different tasks, is the typical benchmark for Knowledge Distillation (KD) in NLP[4]. We also evaluated the proposed approach on the SQUAD.V1 and the IMDB datasets. We repeated the experiments for 5 random seeds and we reported the average (Table 3 and Table 4) or the average/standard deviation (Table 1 and Table 2).
>
> [1] Patient Knowledge Distillation for BERT Model Compression, S. Sun and Yu Cheng and Zhe Gan and Jingjing Liu, EMNLP2019.\
> [2] Knowledge distillation via softmax regression representation learning, Jing Yang, Brais Martinez, Adrian Bulat, Georgios Tzimiropoulos. ICLR2021.\
> [3] TinyBERT: Distilling BERT for Natural Language Understanding. Xiaoqi Jiao, Yichun Yin, Lifeng Shang, Xin Jiang, Xiao Chen, Linlin Li, Fang Wang, Qun Liu. EMNLP Findings 2020.\
> [4] BERT Learns to Teach: Knowledge Distillation with Meta Learning. Wangchunshu Zhou, Canwen Xu, Julian McAuley. ACL2021.

---

### Meta-Review · Area_Chair_2fi1 · 2023-09-18

**Recommendation:** 3

**Metareview:**

This work proposes a new distillation loss which maximizes the cross correlation between individual features at pre-logit activations. Authors apply their method on various BERT models fine-tuned on downstream tasks. Reviewers agree on the novelty of the method and find the results promising. Authors also provide further ablations and comparison to related work during the rebuttal.

---

### Decision · Program_Chairs · 2023-10-07

**Decision:**

Accept-Findings

**Comment:**

This work proposes a new distillation loss which maximizes the cross correlation between individual features at pre-logit activations. Authors apply their method on various BERT models fine-tuned on downstream tasks. Reviewers agree on the novelty of the method and find the results promising. Authors also provide further ablations and comparison to related work during the rebuttal.